# Effects of Sulfide Input on Arsenate Bioreduction and Its Reduction Product Formation in Sulfidic Groundwater

**DOI:** 10.3390/ijerph192416987

**Published:** 2022-12-17

**Authors:** Yang Yang, Xi Xie, Mengna Chen, Zuoming Xie, Jia Wang

**Affiliations:** 1Hubei Key Laboratory of Yangtze Catchment Environmental Aquatic Science, School of Environmental Studies, China University of Geosciences, Wuhan 430074, China; 2School of Chemistry and Chemical Engineering, Shihezi University, Shihezi 832003, China; 3State Key Laboratory of Biogeology and Environmental Geology, China University of Geosciences, Wuhan 430074, China; 4Changjiang River Scientific Research Institute, Wuhan 430014, China

**Keywords:** arsenic-reducing bacteria, groundwater, orpiment, sulfide, thioarsenite

## Abstract

Microbes have important impacts on the mobilization of arsenic in groundwater. To study the effects of sulfide on As(V) bioreduction in sulfidic groundwater, Citrobacter sp. JH012-1 isolated from sediments in the Jianghan Plain was used in a microcosm experiment. The results showed that sulfide significantly enhanced As(V) bioreduction as an additional electron donor. The reduction rates of As(V) were 21.8%, 34.5%, 73.6% and 85.9% under 0, 15, 75 and 150 µM sulfide inputting, respectively. The main products of As(V) bioreduction were thioarsenite and orpiment and the concentration of thioarsenite reached to 5.5 and 7.1 µM in the solution with the initial 75 and 150 µM sulfide, respectively. However, under 0 and 15 µM sulfide inputting, the dominant product was arsenite with no thioarsenite accumulation. The decrease in pH enhanced the bioreduction of As(V) and promoted the formation of thioarsenite and orpiment. In addition, the percentage of thioarsenite in total arsenic decreased with the decrease in the ratio of sulfur to arsenic, indicating that the formation of thioarsenite was limited by the concentration of initial sulfide. Therefore, the presence of sulfide had a significant effect on the transformation of arsenic in groundwater. This study provides new insights into the bioreduction of As(V) and the formation of thioarsenite in sulfidic groundwater.

## 1. Introduction

Arsenic is a common metalloid element in nature and is widely distributed in air, water, soil and sediment [1,2]. As a carcinogen, arsenic is posing a potential threat to human health [3]. It was reported that the concentration of As in groundwater in the Jianghan Plain ranges from 31.5 to 1470 μg/L, far exceeding WHO limit of 10 μg/L [4]. Oxidative dissolution of As-rich pyrite and reductive dissolution of As-rich ferric hydroxide are considered to be the main reasons for the aggravation of arsenic pollution in groundwater [5,6].

Arsenic concentration in groundwater is affected by microbial activities, such as the reduction of As (V) to As(III) by arsenic-reducing bacteria, resulting in the release of arsenic in sediments [7]. Arsenate-reducing bacteria include dissimilated arsenate-reducing bacteria and cytoplasmic arsenate-reducing bacteria [8]. Dissimilated arsenate-reducing bacteria are regulated by the *arr* operon and obtain energy by reducing As(V) to support cell growth and biochemical function [9]. Cytoplasmic arsenic-reducing bacteria mediated by *ars* operon can reduce intracellular As(V) to As(III) and expel As(III) from cells to reduce the toxicity of As(V) [10]. Some bacteria have both *arrA* and *arsC* genes, such as *Shewanella* sp. ANA-3 and *Bacillus* XZM [11,12].

As another common element in groundwater, sulfur species usually coexist with As, and their interaction is of great significance in the biogeochemical cycle [13]. The free sulfide in groundwater comes from the activity of microorganisms. More specifically, sulfate-reducing bacteria (SRB) and elemental sulfur-reducing bacteria (S0RB) can reduce sulfate and elemental sulfur in groundwater to sulfide [14,15]. In the presence of free sulfide, As-bearing iron oxide minerals will be reduced by sulfide and release adsorbed arsenic. The formation of soluble thioarsenic species may enhance the mobility of arsenic in excessive sulfide groundwater [16,17]. In addition, sulfide can combine with As(III) to form insoluble As–S compounds (orpiment, realgar) to effectually remove arsenic [18,19,20]. Sulfur-bearing iron minerals such as pyrite, mackinawite and arsenopyrite can also immobilize arsenic [21,22]. However, these arsenic removal methods are limited by environmental conditions [18].

In addition to obtaining energy from organic matter, dissimilatory arsenate-reducing bacteria can use inorganic matter such as hydrogen and sulfide as electron donors [23,24]. Hoeft et al. first isolated an alienation arsenic-reducing bacteria MLMS-1 with sulfide as an electron donor from a sample obtained at the bottom of Mono Lake [25]. Hollibaugh et al. also demonstrated that an electron transfer took place between sulfide and As(V) [26]. The chemical reaction between sulfide and As(V) is kinetically feasible, but in a neutral groundwater environment, this reaction cannot easily occur [27]. Interestingly, the reaction of As(III) and sulfide means that the thioarsenic species and orpiment may be formed during As(V) bioreduction in a sulfidic environment [20,28]. Moreover, the ratio of sulfur to arsenic as well as the pH in groundwater are important factors affecting the reaction between As(III) and sulfide [29].

Previous studies have focused on the effect of sulfide on As(V) bioreduction, ignoring its impact on the reduction products of As(V) [25,26]. In this study, a new arsenate-reducing bacterium *Citrobacter* sp. JH012-1 from the sediments of the Jianghan Plain were used to explore the effect of sulfide on As(V) bioreduction and the formation of thioarsenic species in sulfidic groundwater. The objectives of this study were to: (i) explore the product of the reaction of As(III) with sulfide after As(V) bioreduction; (ii) determine the effect of initial sulfide concentration and pH on As(V) bioreduction; and (iii) explore the effect of the ratio of sulfur to arsenic on As(V) bioreduction products.

## 2. Materials and Methods

### 2.1. Collection of Sediment Samples

Sediment samples were collected from a depth interval of 28.6–28.8 m in a 40-m deep borehole in Shayang County (30°45′ N, 112°32′ E), Hubei Province. Samples packed in polyethylene bags were vacuumed and stored in an incubator filled with dry ice, and then transported to the laboratory. Sediment samples were stored in a −20 °C refrigerator for subsequent bacterial isolation.

### 2.2. Bacterial Isolation and Identification

To obtain an indigenous arsenate-reducing bacteria, one gram of sediment sample was diluted with 9 mL of sterile water and then spread on the selective medium plate in an anaerobic glove box. The selective medium included Na_2_S_2_O_3_∙5H_2_O (5.0 g/L), KNO_3_ (2.0 g/L), KH_2_PO_4_ (2.0 g/L), NaHCO_3_ (1.0 g/L), MgCl_2_∙6H_2_O (0.5 g/L), NH_4_Cl (0.5 g/L), FeSO_4_∙7H_2_O (0.01 g/L), and a pH of 7.0. The selective medium was added with 13.3 µM As(V). This arsenic concentration level was chosen according to the average arsenic concentration in the groundwater at the Jianghan Plain [4]. The plates were incubated in a dark, nitrogen-filled incubator at 30 °C for 2–3 d. Colonies with different morphological characteristics were selected and inoculated in an anaerobic bottle containing 100 mL of selective medium and incubated in a shaking table at 30 °C and 180 r/min for 3–4 d. The colonies with different morphologies were separated by spreading on the selective medium plate three consecutive times. Strain JH012-1 was found to have the best As(V)-reducing performance and a fast growth rate.

The nucleotide sequence of the isolated bacterial 16S rRNA gene was amplified by using a universal primer designed by Sangon Biotech [12]. The universal primer sequences were 5′AGAGTTTGATCMTGGCTCAG and 5′GGTTACCTTGTTACGACTT. PCR amplification was performed as described in Xie et al. [30]. Bacterial 16S rRNA gene sequences were analyzed using a BLAST server and MEGA 7.0 software package [30]. Strain JH012-1 has the gene bank accession number MZ227386 of SUB9679906 *Citrobacter*.

### 2.3. Batch Experiments

To simulate the actual groundwater environment, the anoxic minimal salt medium (AMSM) used in the batch experiment contained 6.7 mM KCl, 1 mM CaCl_2_, 2 mM NH_4_Cl, 3 mM MgCl_2_∙6H_2_O, 17 mM NaCl, 0.1 mM KH_2_PO_4_, 3.3 mM glucose and 0.01% yeast extract [12]. The reagents used were analytical grade and all solutions were prepared with ultra-pure (UP) water. All experiments were carried out under strict anaerobic conditions and repeated three times to eliminate errors.

#### 2.3.1. Bioreduction Products of Arsenate in the Presence of Sulfide

The concentration of As(V) in AMSM was controlled 
to 13.3 µM, which was based on the arsenic concentration range from 0.4 to 19.6 
µM in the Jianghan Plain [4]. The initial 
concentration of sulfide was 150 µM to ensure excess sulfide. Forty mM Bis 
[2-Hydroxyethyl] imino Tris-(Hydroxymethyl)-methane (Bis-Tris) was used to 
stabilize the pH of the medium at 7. Both the anaerobic bottle and the culture 
medium were sterilized by autoclaving at 121 °C for 20 min for the following 
experiments. About 100 mL of the medium was dispensed into the anaerobic bottle 
and the solution was inflated with high-purity nitrogen for 20 min to remove 
the remaining oxygen. After the aeration, the anaerobic bottles with 5 mL of 
exponentially-growing cultures were sealed with a rubber stopper and incubated. 
The initial cell concentration in the anaerobic bottles was 5 × 10^6^ 
CFU/mL. A sterile control group was set in the same way. The sealed anaerobic 
bottle was then incubated in a biological incubator at 32 °C in the dark. At 
the end of the experiment, liquid samples and precipitates were collected. The 
liquid samples were filtered and solid samples were freeze dried for Raman 
analysis. A Raman spectrometer (Roman, enishaw inVia) with a wavelength of 785 
nm was used to identify the thioarsenic species in the solutions and 
precipitates.

#### 2.3.2. Effect of Initial Sulfide Concentration and pH on Arsenate Bioreduction

Forty mM Bis-Tris was added to the medium to stabilize the pH at 7. The concentration of As(V) in medium was maintained at 13.3 µM. The initial sulfide concentrations were set at 0, 15, 75 and 150 µM to study the effect of different initial sulfide concentrations on As(V) bioreduction. The sulfide concentration series was selected due to the low free sulfide concentration in groundwater. To investigate the effect of pH on As(V) bioreduction, the pH was adjusted to 6 and 8 in the control experiments.

#### 2.3.3. Effect of the Ratio of Sulfur to Arsenic on Arsenate Bioreduction Products

Different sulfur to arsenic ratios were set to explore their effect on the As(V) bioreduction product. The concentration of sulfide in the medium was 0.2 mM and the As(V) concentrations were set to 0.1, 0.2, 0.5 and 1 mM to control the sulfur-arsenic ratio at 2:1, 1:1, 1:2.5 and 1:5. Other treatments were the same as those mentioned in Section 2.3.1.

### 2.4. Chemical Analysis

In batch experiments, samples were collected with a syringe and filtered with 0.22 μm polyethersulfone membrane filter. Concentrations of As(T), As(V) and As(III) in the filtrate were measured by hydride generation atomic fluorescence spectroscopy (HG-AFS; AFS-830, Beijing Jitian Instrument Co., Ltd., Beijing, China). The separation of As(V) and As(III) was achieved by a silica-based strong anion-exchange cartridge [31]. The As–S compounds concentrations were calculated as As(T) minus As(V) and As(III) [27]. Sulfide in filtrate was determined by the methylene blue method [32]. Sulfate was determined using an ion chromatograph (Eco IC, Metrohm, Herisau, Switzerland) with an analytical precision of ±3.0%.

## 3. Results

### 3.1. Phylogenetic Analysis

The strain sequence was blast compared on the ezbicloud website. The top 25 sequences with the highest similarity in the results were selected as the reference sequence. The 16S rRNA gene of *Streptomyces albus* NBRC 13014 from different phylum in the taxonomic hierarchy was used as the outgroup to construct the NJ model tree. The phylogenetic tree was constructed by Kimura-2 and Bootstrap tested 1000 times. Phylogenetic analysis showed that the strains were clustered in the same branch with *Citrobacter werkmanii* NBRC 105721 and *Citrobacter cronae* Tue2 1, and the bot-up value was 51, indicating that the topological structure of the developmental tree was generally stable (Figure 1). In combination with the BLAST results, the similarities of the 16S rRNA gene sequences of strain JH012-1 with *Citrobacter werkmanii* NBRC 105721 and *Citrobacter cronae* Tue2 1 were over 99.5%, suggesting it belongs to the genus *Citrobacter*. Arsenate-reducing bacteria such as the *Citrobacter* sp. strain TSA-1 [33] have been reported. Strain JH012-1 used in this study is considered to be another member of the *Citrobacter* family with arsenate-reducing capacity.

### 3.2. Bioreduction Products of Arsenate in the Presence of Sulfide

The changes of dissolved arsenic and sulfur concentrations in medium during the incubation time are shown in Figure 2. In abiotic control experiments, there was no significant change in the As(V) concentration, indicating that chemical redox reactions between As(V) and sulfide were difficult under neutral conditions. Compared with the abiotic control, the consumption of As(V) and sulfide in biotic incubation increased significantly. The As(V) and sulfide concentration dropped by 85.9% and 34.1% at the end of the incubation (Figure 2a,b). Interestingly, the concentration of As(III) was negligible.

The As(T) concentration in biotic incubation decreased by 28.3%, while the As(T) concentration in the abiotic control was steady. There was a difference between the dissolved As(T) concentration and the sum of As(V) and As(III) concentrations in biotic incubation. Raman spectra of liquid samples (Figure 2c) showed three strong spectral peaks in the region of 380–450 cm^−1^. The clear yellow precipitate can be observed in the anaerobic bottle at the end of the experiment. Raman spectrum analysis of the precipitate showed (Figure 2d) that the solid sample spectrum peak was similar to the orpiment. These results suggest that the decrease in As(T) was due to the formation of orpiment and thioarsenic. The soluble thioarsenic concentrations were calculated by subtracting As(V) and As(III) concentration from the As(T) concentration. At the end of the experiment, a 7.1 µM thioarsenic species formed in biotic incubation experiments, accounting for 53.4% of the initial As(V) concentration.

### 3.3. Effect of Initial Sulfide Concentration and pH on Arsenate Bioreduction

The changes of sulfide concentration under different initial sulfide concentrations are shown in Figure 3. When 15 and 75 µM of sulfide were input, the sulfide concentrations in abiotic incubations showed a slight downward trend and decreased to 13.4 and 58.9 µM, respectively. Compared with the abiotic incubations, the sulfide concentration in biotic incubations decreased more rapidly and reached 7.0 and 49.5 µM at the end of the experiment. The changes in the sulfide concentration with 150 µM of initial sulfide input are shown in Figure 2b. Sulfide consumption in biotic incubations with 15, 75 and 150 µM of sulfide input were 8.0, 25.5 and 51.1 µM, respectively. Figure 4 shows the impact of the initial sulfide concentration on As(V) bioreduction. During the biotic incubation, the As(T) concentration continued to decrease and reached 13.1, 12.0, 10.6 and 9.6 µM in the treatments with 0, 15, 75 and 150 µM of sulfide input (Figure 4a). The changes in As(T) concentration for abiotic controls were negligible. In the biotic incubation without sulfide input, As(V) was reduced slowly and the content of As(V) and As(III) reached 10.4 and 2.6 µM, respectively. The finally As(V) concentration decreased from 10.4 to 8.7, 3.5 and 1.9 µM when the sulfide concentration increased from 0 to 15, 75 and 150 mM, respectively (Figure 4b). The concentration of As(III) in the medium was 3.2 µM when 15 µM sulfide was input. Curiously, as the sulfide concentration was further increased from 15 µM to 75 µM and 150 µM, the As(III) concentration in solution instead decreased from 3.2 µM to 1.7 and 0.6 µM (Figure 4c), respectively. In addition, the accumulation of As–S compounds only occurred when the sulfide concentrations were 75 and 150 µM, and reached 5.5 and 7.1 µM at the end of the experiment, respectively (Figure 4d). These results show that the increase in the initial concentration of sulfide is beneficial to the bioreduction of As(V) and promotes the formation of thioarsenic species.

Figure 5 shows the changes in the concentrations of different arsenic species at pH values of 6, 7 and 8. The concentration of As(T) increased with the increase in pH, and the final concentrations at pH 6, 7 and 8 were 8.5, 9.6 and 10.7 µM, respectively (Figure 5a). The reduction rate of As(V) was negatively correlated with the increase in pH, and the final reduction rate of As(V) at pH 6, 7 and 8 was 89.5%, 85.9% and 70.8%, respectively (Figure 5b). The increase in As(III) concentration was not significant under the three pH conditions, but the concentration of As(III) at pH 6 was slightly higher than that at pH 7 and pH 8 (Figure 5c). In the treatment at pH 6, As–S compounds reached their peak on the third day and then declined slightly while the As–S compound concentration continued to increase at pH 7 and 8. The final concentrations of As–S compounds were 5.8, 7.1 and 6.3 µM at pH 6, 7 and 8, respectively (Figure 5d). These observations indicate that the decrease in pH promotes the bioreduction of As(V) and the formation of realgar. Interestingly, although the rate of thioarsenic species formation was accelerated at pH 6, the thioarsenic content at pH 6 was lower than that at pH 7 and 8.

### 3.4. Effect of the Ratio of Sulfur to Arsenic on Arsenate Bioreduction Products

Figure 6 shows the change in sulfide concentration in the medium. In the treatment with 0.1 mM As(V) input, the sulfide was rapidly consumed in the first two days and remained stable after reaching 0.1 mM. When the As(V) concentration increased to 0.2, 0.5 and 1 mM, the content of sulfide decreased to 0.08, 0.04 and 0.03 mM, respectively. This result indicates that the increase in As(V) concentration increases the consumption of sulfide.

Figure 7 shows the effect of the ratio of sulfur to arsenic on As(V) bioreduction. When the ratio of sulfur to arsenic was gradually decreased from 2:1 to 1:1, 1:2.5 and 1:5, the removal rate of As(T) in the solution dropped from 28.7% to 25.5%, 17.9% and 16.7%, and the reduction rate of As(V) also dropped from 85.0% to 84.5%, 83.7% and 81.9%. Interestingly, although the reduction rate of As(V) decreased with a decreasing sulfur-arsenic ratio, the percentage content of As(III) increased gradually, accounting for 16.8%, 33.8%, 53.8% and 65.7% of the initial arsenic concentration. The accumulation of arsenic-sulfur compounds was 35.6%, 25.1%, 12.0% and 3.5% under four different sulfur to arsenic ratios. These results suggest that the decrease in the ratio of sulfur to arsenic leads to the gradual transformation of As(V) bioreduction products, and the main products are from arsenic-sulfur compounds to As(III).

## 4. Discussion

Our results indicate that sulfide significantly affects the bioreduction and reduction products of As(V). The increase in the initial sulfide concentration in the medium promoted the As(V) bioreduction, and soluble As–S compounds and orpiment were formed in the medium with 150 µM sulfide. No arsenic changes were observed in the sterile control experiments. The formation of As–S compounds was closely related to the initial concentration of sulfide in the solution [34,35]. In this study, As–S compounds were formed only when the concentration of sulfide exceeded 75 µM. In addition, the ratio of sulfur to arsenic was another important interfering factor for As(V) bioreduction products.

During the bioreduction of As(V), the accumulation of As(III) in the solution was not synchronized with the reduction of As(V), which was due to the reaction of As(III) with excess sulfide. Many laboratory and field studies have shown that the interactions of As(III) and sulfide were the dominant pathway to form As–S compounds [26,28,36]. There was no change in As(V) concentration in the sterile control, confirming that sulfide cannot react directly with As(V) to form thioarsenate. Therefore, the As–S compounds produced in the biotic incubation experiment may be thioarsenite. In this study, Raman spectral data proved the presence of thioarsenite in the solution. The formation of three strong spectral peaks in the region 380–450 cm^−1^ can be attributed to the As–S bonds [37,38,39]. Wood et al. found that there were at least 6–8 different thioarsenic species in the region of 300–450 cm^−1^ [38]. Moreover, there were many obvious peaks between the region of 750–900 cm^−1^ due to the As(III) species [38]. Thioarsenite has multiple protonation states. The abundance of each state depends on the sulfur to arsenic ratio in sulfidic environments, and trithioarsenite is the dominant thioarsenic species in the presence of excess sulfide [28,29,40]. Price et al. revealed the pathway in which As(III) reacted with excess sulfide to form trithioarsenite under reducing conditions, as shown in reaction 1 [41]. However, due to the high instability of thioarsenite under aerobic conditions, and that the concentration of arsenic required by any measurement method is about one order of magnitude higher than that in this experiment, we cannot deliver proof for the presence of trithioarsenite. Thioarsenite is easily oxidized to form thioarsenate [42]. Considering that all experiments in this study were under anaerobic conditions, the oxidation reaction was negligible. In addition to thioarsenite, Raman spectrum analysis of the precipitate showed that orpiment was another product of the reaction between As(III) and sulfide [27,38]. Reaction 2 shows the formation of orpiment [18,19,20]. Moreover, some investigations provided new insight that the presence of excessive sulfide may transform insoluble As–S precipitates into soluble thioarsenic species [29,40].
(1)$$H_{3}{\mathrm{AsO}}_{3}+3S^{2-}+6H^{+}\to H_{3}{\mathrm{AsS}}_{3}+3H_{2}O$$
(2)$$2H_{3}{\mathrm{AsO}}_{3}+3S^{2-}+6H^{+}\to {\mathrm{As}}_{2}S_{3}↓+6H_{2}O$$

The increase in sulfide concentration and the decrease in pH accelerated the bioreduction of As(V) and promoted the formation of thioarsenite and orpiment [18,36]. Arsenate-reducing bacteria used sulfide as an additional electron donor to reduce As(V) [25,43]. The reduction of As(V) by arsenic-reducing bacteria includes two pathways: cytoplasmic reduction and dissimilatory reduction [23]. Cytoplasmic reduction means that after As(V) enters the cell through the phosphate transporter, it is reduced to As(III) and excreted out of the cell under the action of the *ars* operon [7,23]. The dissimilatory reduction mechanism is that bacteria can use As(V) as terminal electron acceptor, combining the reduction of As(V) with the oxidation of organic (glucose, acetate, pyruvate) and inorganic (sulfide and hydrogen) substrates, thus providing energy for the growth of bacteria [23,25]. Therefore, the input of exogenous electron donors can significantly promote the bioreduction of As(V) [43]. Bacteria can also be used as an electron shuttle to accelerate the redox reaction of sulfide and As(V) under neutral conditions [44]. From a kinetic point of view, more sulfide and protons are conducive to the progress of reactions 1 and 2 which can accelerate As(III) to reach saturation with respect to amorphous orpiment [45]. No accumulation of thioarsenite was observed when 15 µM sulfide was input. When the initial sulfide concentration increased to 75 or 150 µM, As(III) was converted to thioarsenite, suggesting that the formation of thioarsenite was limited by the sulfide concentration [35]. This was related to the ligand exchange mechanism for thioarsenite formation, a process in which the -OH groups in As(III) were gradually replaced by -SH [46]. Because of the competitive dissociation between -SH and -OH, As(III) will be easier to combine with -SH to form thioarsenite, whether the pH decreases or the sulfide concentration increases [35]. Interestingly, the content of thioarsenite at pH 6 was lower than at pH 7 and 8. This is due to the hydrolysis of thioarsenite to re-form As(III) [42,47]. In addition to pH and the concentration of sulfide and arsenic, As(V) bioreduction efficiency was linked to other factors, such as iron and nitrate [48,49].

The consumption of sulfide was due to the bioreduction of As(V) and the formation of thioarsenic species. The increase in the concentration of As(V) increased the content of As(III) produced by As(V) bioreduction, and the increase in As(III) content subsequently promoted the formation of thioarsenic species [42,50]. With the increase in As(V) concentrations, the concentration of thioarsenite increased slightly, but the increase in As(III) concentration was more significant. The main product of As(V) bioreduction is gradually converted from thiosarsenate to As(III) when the ratio of sulfur to arsenic decreases because the sulfide is not enough to support the formation of more thiosarsenite. In addition, the percentage of thioarsenite in total arsenic decreased with the decrease in the ratio of sulfur to arsenic, proving that the formation of thioarsenite was limited in the free sulfide solution [35].

Previous studies have shown that the sulfide produced by sulfate reduction significantly affects the migration and transformation of arsenic in groundwater [15,18]. The chemical reaction of sulfide with As(V) was kinetically unfavorable in the solution at pH 7 [27]. Therefore, the effect of sulfide on As(V) bioreduction should be given more attention in neutral groundwater environments. The reduction rate of As(V) by strain JH012-1 was affected by the initial sulfide concentration and pH. As an important product of As(V) bioreduction in a sulfidic system, thioarsenite was only formed in the presence of excess sulfide [35]. The main reduction product of As(V) transformed from thioarsenite to As(III) with the decrease in the ratio of sulfur to arsenic. In studying the effect of sulfide on As(V) bioreduction and reduction products, it is helpful to understand the pathway by which sulfide affects arsenic transformation in groundwater to better clarify arsenic mobilization in sulfidic groundwater. This also provides a more detailed theoretical basis for microbial remediation in a high-arsenic groundwater environment.

## 5. Conclusions

The purpose of this study was to explore the effect of sulfide input on As(V) bioreduction and the reduction products. The interaction between sulfur and arsenic can result in arsenic redistribution in groundwater. Since the chemical redox reaction between sulfide and As(V) was difficult in a neutral anaerobic environment, the effect of sulfide on As(V) bioreduction should be considered. The main products of As(V) bioreduction were orpiment and thioarsenite instead of As(III) when sulfide was input. Interestingly, thioarsenite accumulation was only observed at initial sulfide concentrations above 75 µM. The increase in the initial sulfide concentration and the decrease in pH enhanced the bioreduction of As(V) and accelerated the formation of orpiment and thioarsenite. Moreover, the formation of thioarsenite was limited by sulfide concentration when the ratio of sulfur to arsenic decreased in groundwater. The input of sulfide promotes the bioreduction of As(V) and causes the formation of thioarsenite and orpiment, which is the primary way that sulfide affects arsenic transformation in groundwater.

## Figures and Tables

**Figure 1 ijerph-19-16987-f001:**
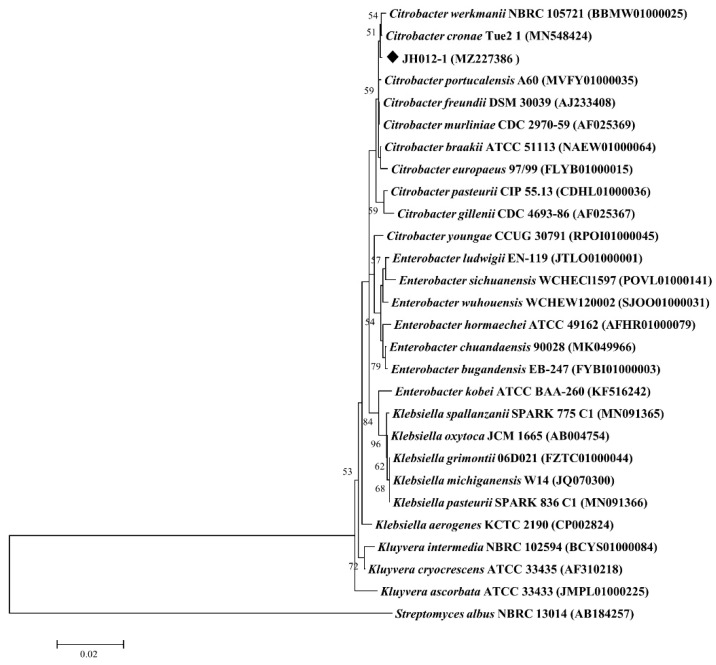
Neighbor-joining phylogenetic tree for the strain JH012-1 based on 16S rRNA sequences. The strain used in this study was marked with black diamond squares.

**Figure 2 ijerph-19-16987-f002:**
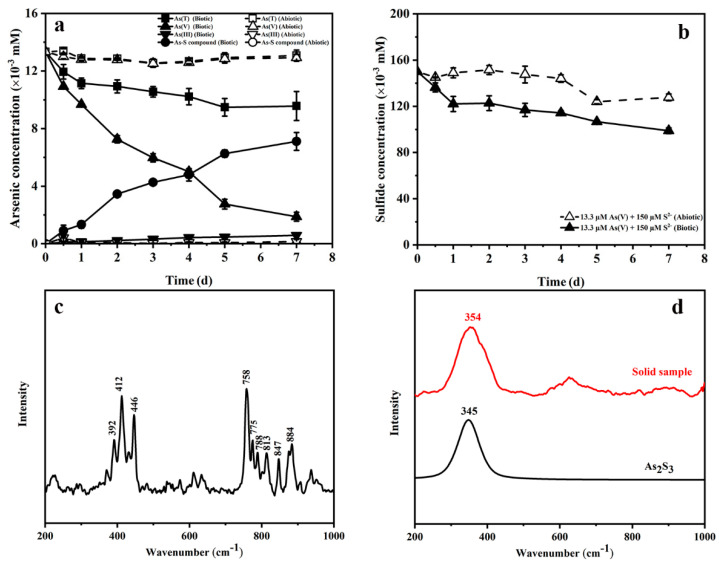
Changes in arsenic (**a**) and sulfur (**b**) concentration in the medium with or without strain JH012-1. Raman spectra of the solution (**c**) and the precipitation (**d**) after 144 h of reaction.

**Figure 3 ijerph-19-16987-f003:**
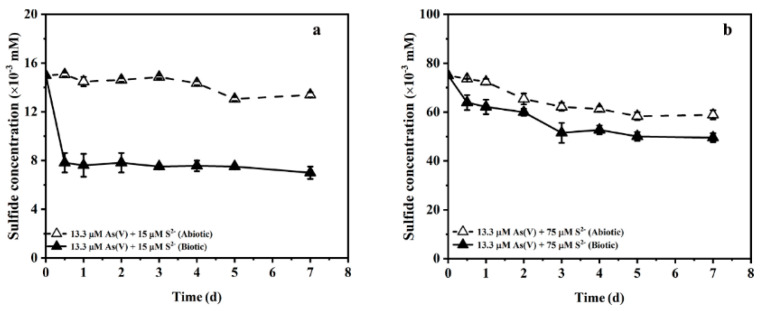
Changes in sulfide concentration in the medium with 15 mM (**a**) or 75 mM (**b**) initial sulfide input.

**Figure 4 ijerph-19-16987-f004:**
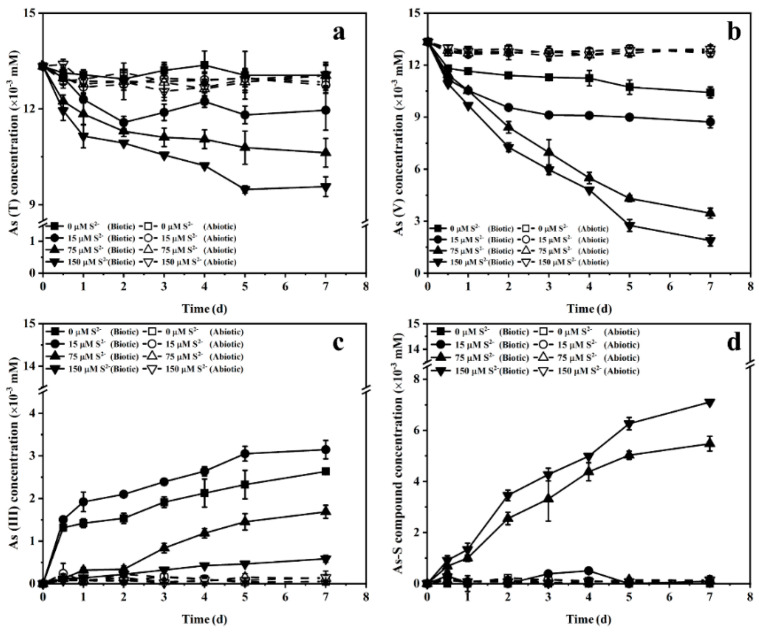
The concentrations of As(T) (**a**), As(V) (**b**), As(III) (**c**) and As–S (**d**) in the medium with or without strain JH012-1 under different initial sulfide concentrations.

**Figure 5 ijerph-19-16987-f005:**
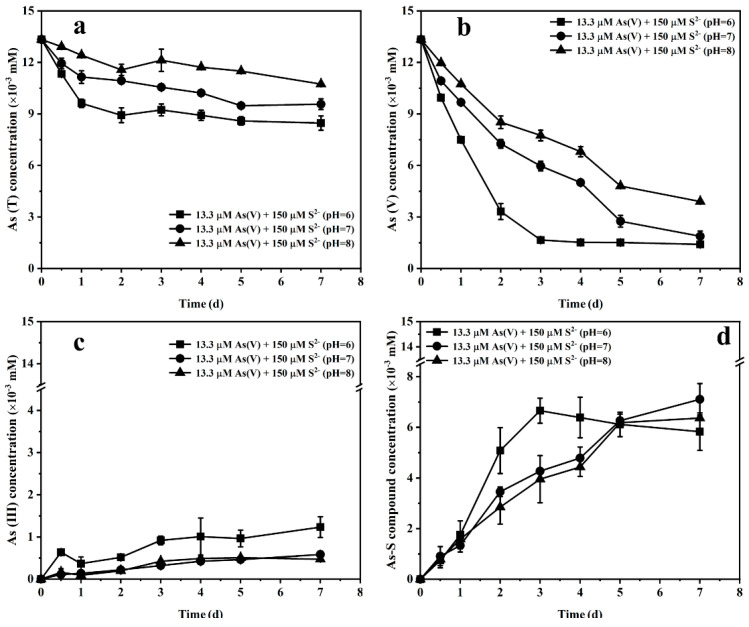
The concentrations of As(T) (**a**), As(V) (**b**), As(III) (**c**) and As–S (**d**) in the medium with strain JH012-1 at pH 6, 7 and 8.

**Figure 6 ijerph-19-16987-f006:**
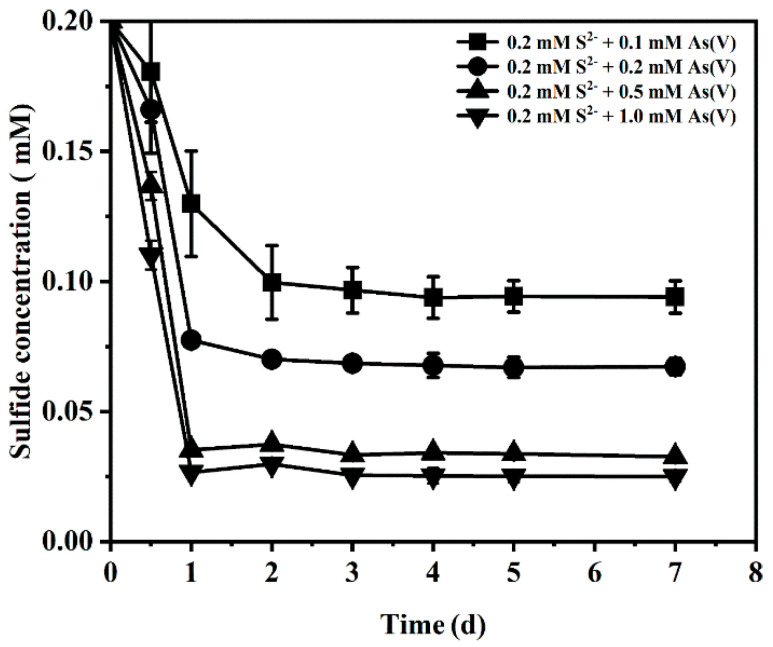
The change in sulfide concentration in the medium under different sulfur-arsenic ratio conditions.

**Figure 7 ijerph-19-16987-f007:**
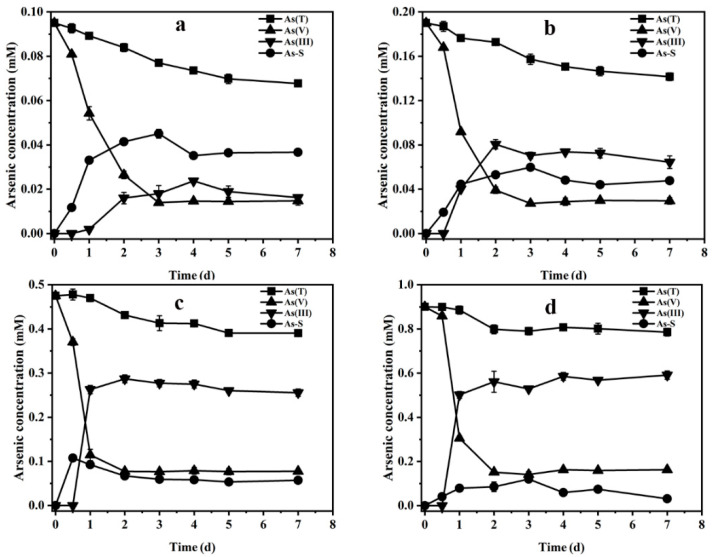
Changes in arsenic concentration in the medium when the ratio of sulfur to arsenic was 2:1 (**a**), 1:1 (**b**), 1:2.5 (**c**) and 1:5 (**d**).

## Data Availability

Not applicable.

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
