# Peer review of "Effects of Sulfide Input on Arsenate Bioreduction and Its Reduction Product Formation in Sulfidic Groundwater"

_ijerph, 2022, doi:10.3390/ijerph192416987_

Round 1
Reviewer 1 Report
This manuscript aimed to investigate the changes of As(V) bio-reduction products after sulfide input, and the effects of sulfide input on the bio-reduction rate of As(V) were compared under different sulfur to arsenic ratios and pH. The data obtained from the experiments were valuable for solidification of the understanding on the reduction mechanism and stability of arsenate in sulfidic groundwater. The subject of this manuscript was of interest and fitted well with the scope of International Journal of Environmental Research and Public Health.
This study mainly addressed the interaction mechanism between sulfide, arsenate and microorganisms in sulfidic groundwater, and the bioreduction rate and reaction products of sulfide to arsenate were studied in detail under different pH conditions and sulfur-arsenic ratios, providing new insights into orpiment and thioarsenite formation in sulfidic groundwater environments. I think this topic is much relevant in the field. Previous studies have proposed the abiotic reduction mechanism of sulfide to arsenate and the promotion of sulfide to arsenate reduction by dissimilatory arsenic-reducing bacteria. On this basis, this study discussed the possibility of abiotic reduction of sulfide and arsenate in a neutral groundwater environment, and the effects of pH and sulfur-arsenic ratio on the interaction between sulfide, arsenate and microorganisms were discussed in detail. New insights into the formation conditions and interconversion of thioarsenic species in sulfidic groundwater environments. The characterization of the thioarsenite species in this study is weak and the content of soluble thioarsenite is not directly measured. Conclusions are consistent with the evidence and arguments presented, and they address the main question proposed.
Overall, the manuscript was well organized and easy to follow, a minor revision was suggested before its acceptance.
Specific comments:
1. Title. Sulfidic groundwater means the actual groundwater, however there is no description of groundwater extraction in the full text.
2. Line 23. What does free sulfide mean? If it is initial solution sulfur, it is best to expressed by initial sulfide concentration, and if it is the remaining unbound sulfur in the system after the reaction, please specify.
3. Line 81. What’s this? I didn't see the content of this study. Please modify.
4. Line 102. Correct the title.
5. Line 112. Please explain the first abbreviations.
6. Line 117. What is the initial cell concentration used in the experiments?
7. Line 132. Change " its " to " their ".
8. Line 173. Space between "28.3 %".
9. Line 176. What’s the meaning of these wavenumbers?
10. Line 179. The decrease in As(T) concentration is due to orpiment or thioarsenic or both of them, please summarize this paragraph more clearly.
11. Line 269. Change " which due to " to " which was due to "
12. Line 339. Your medium is self-configured and it is not appropriate to express it in groundwater.
13. The Figure legends are not clear enough.
Reviewer 2 Report
Effects of sulfide on As(V) bioreduction in sulfidic groundwater were studied in this manuscript. It was proved that the presence of sulfide had a significant effect on transformation of arsenic in groundwater. This study provided new insights into the bioreduction of As(V) and the formation of thioarsenite in sulfidic groundwater. In my opinion, this work is very interesting and worthy to be accepted after minor revision.
1. What is the real function of sulfide addition should be mentioned in the abstract and should be discussed deeply in the discussion section.
2. The microbial strain should be paid more attention in the whole process, it should be considered in the the bioreduction of As(V). This should be discussed in detail to strengthen microbial function.
Author Response
Specific comments:
1. What is the real function of sulfide addition should be mentioned in the abstract and should be discussed deeply in the discussion section.
Response: Thank you for your suggestion. Sulfide can significantly enhanced As(V) bioreduction as an additional electron donor. On the one hand, arsenate-reducing bacteria can use sulfide as an additional electron donor to reduce As(V). On the other hand, bacteria can be used as electron shuttle to accelerate the redox reaction of sulfide and As(V) under neutral condition. Please see the modification in L 16, L304-305 and L 314-316
2. The microbial strain should be paid more attention in the whole process, it should be considered in the the bioreduction of As(V). This should be discussed in detail to strengthen microbial function.
Response: Thank you for your suggestion. Microbial activity plays an important role in the As(V) bioreduction process. The mechanisms of As(V) reduction by arsenic reducing bacteria have been added in the revised manuscript to explain the role of sulfide in promoting arsenate bioreduction. Please see the modification in L 306-314.